# Research Progress and Application of Single-Atom Catalysts: A Review

**DOI:** 10.3390/molecules26216501

**Published:** 2021-10-28

**Authors:** He He, Hudson Haocheng Wang, Junjian Liu, Xujun Liu, Weizun Li, Yannan Wang

**Affiliations:** 1National & Local Joint Engineering Research Center on Biomass Resource Utilization, College of Environmental Science and Engineering, Nankai University, Tianjin 300071, China; 2120200533@mail.nankai.edu.cn (H.H.); liujunjian@mail.nankai.edu.cn (J.L.); wangyannan@nankai.edu.cn (Y.W.); 2Indian Springs School, Pelham, AL 35124, USA; hudson.wang@indiansprings.org; 3State Key Laboratory of Molecular Engineering of Polymers, Fudan University, Shanghai 200433, China; liuxujun@fudan.edu.cn

**Keywords:** single-atom catalyst, preparation, characterization, application

## Abstract

Due to excellent performance properties such as strong activity and high selectivity, single-atom catalysts have been widely used in various catalytic reactions. Exploring the application of single-atom catalysts and elucidating their reaction mechanism has become a hot area of research. This article first introduces the structure and characteristics of single-atom catalysts, and then reviews recent preparation methods, characterization techniques, and applications of single-atom catalysts, including their application potential in electrochemistry and photocatalytic reactions. Finally, application prospects and future development directions of single-atom catalysts are outlined.

## 1. Introduction

In recent years, human society has faced severe energy shortages. One of the most effective ways to alleviate the energy crisis is to make efficient and reasonable use of existing fossil resources on the earth [1]. To a large extent, the activity of a catalyst plays a decisive role in the efficient conversion of existing fossil energy into clean energy, promotes the development of industry, and improves living standards [2,3,4].

In the past few decades, researchers have conducted in-depth exploration into and technological innovation on the development of single-atom catalysts. The field of single-atom catalysts has become very popular field. In 2011, Zhang et al. [5] prepared a single-atom Pt_1_/FeOx catalyst and proposed the concept of single-atom catalysis for the first time. A dispersed Pt single-atom state was observed by XAFS, spherical aberration electron microscopy, infrared, and other characterization techniques. The catalyst has stronger activity and higher stability and selectivity in a CO oxidation reaction. The loading is 0.17 wt% and it has a specific reaction rate of 0.435 molcoh^−1^ 21 gPt^−1^ at a reaction temperature of 27 °C. This research laid the foundation for the development of single-atom catalysts. In 2012, the Charles Sykes research group [6] heated Pd in a vacuum environment, so that the evaporated metal Pd single atom was captured by the Cu(111) surface, and the existence of an isolated Pd single atom was proved using scanning tunneling microscopy. This catalytic system showed high selectivity in the hydrogenation reaction of organic compounds. The hydrogenation reaction of styrene and Ha resulted in ethylbenzene, which was desorbed at 260 K with a conversion rate of 13% and a selectivity higher than 95%. This research promoted the development of the field of single-atom catalysis and proved the special advantages of single atoms. In 2016, Zheng et al. [7] used a simple photochemical method to prepare a stable and well-dispersed Pd/TiO_2_ single-atom catalyst at room temperature, with a Pd loading as high as 1.5%. It had high activity and stability in the hydrogenation reaction of C=C and C=O double bonds. After 20 cycles, the activity of the catalyst did not decrease significantly. These groundbreaking studies gave strong momentum to the field of single-atom catalysis. Since then, researchers have devoted themselves to preparing single-atom catalysts with more variety and complexity, and applying them to a series of catalytic reactions.

A single-atom catalyst (SAC) (Figure 1) is a noble metal or non-noble metal that is uniformly dispersed and fixed on support material with an isolated single atom as its active center. A supported single-atom catalyst has high activity. In the past ten years, people have conducted extensive research and placed attention on the properties of the active centers of single-atom catalysts and the catalytic reaction mechanism. In traditional supported catalysts, only a few metal active components play a catalytic role in the reaction process, and the cost of large-load precious metals is too high, which is not conducive to large-scale development and use. Single-atom catalysts have huge application potential in the rational use of metal resources and the realization of an atomic economy [8]. Because an SAC has a special geometric configuration and electronic structure that is different from traditional catalysts, in particular the valence shell electronic structure of metal atoms, its physical and chemical properties are unique and novel. It exhibits excellent catalytic activity in a series of catalytic reactions such as photocatalysis, electrocatalysis, and thermal catalysis [9]. A single-atom catalyst with low-temperature activity and thermal stability can also be used for the purification of industrial motor vehicle exhaust [10]. In this review, the latest research progress of single-atom catalysts is summarized. The core problem of the synthesis of single-atom catalysts is discussed. The applications of single-atom catalysts in electrochemistry and photocatalysis are introduced, and the structure–activity relationship between the structure of single-atom catalysts and their catalytic performance is discussed.

## 2. The Structure and Characteristics of Single-Atom Catalysts

### 2.1. Structure and Activity of a Single-Atom Catalyst

The isolated metal atoms in the single-atom catalyst are fixed on the support through the strong interaction between the metal and the support or through the formation of a stable chemical bond with the support material. According to different interactions, they are divided into different types of catalysts, mainly including metal single atoms anchored on the surface of metal oxides, metals, carbon-based materials, metal–organic framework materials (MOFs), covalent organic framework (COF) materials, and composite materials. The construction of a single-atom catalyst with a controllable coordination environment and close to the photoactive site in the MOF material can effectively promote the charge transfer during the photocatalytic reaction and enhance the photocatalytic performance [11,12,13,14,15,16,17].

The particle size of the single-atom catalyst determines its high catalytic efficiency. In the evolution of catalytic materials from bulk materials, nanoparticles, and sub-nano clusters to single metal atoms, the surface free energy of metals increases sharply as the particle size decreases. The stronger the interaction between the active site and the support, the higher the activity of the catalyst. When the metal is fixed on the carrier in the form of a single atom, its surface free energy reaches the maximum and the catalyst activity is the strongest. With the rapid development and advancement of characterization technology, researchers have discovered that the active sites of single-atom catalysts are usually unsaturated coordination atoms on the surface of the carrier. By adjusting the size, morphology, and crystal face of the nanocrystals, the distribution and structure of the atoms on the catalyst surface can be changed, therefore enhancing its catalytic activity. When the size of nanocrystal is reduced to the form of clusters of atoms and single atoms, its energy level structure and electronic structure will also be completely changed. Because of its unique structural characteristics, single-atom catalysts show higher activity, stability, and selectivity than traditional nanocatalysts, and efficiently promote the progress of catalytic reactions [18].

### 2.2. Performance Characteristics of Single-Atom Catalysts

Compared with traditional nanoparticle catalysts, single-atom catalysts use metal atoms to the greatest extent, have uniform and adjustable metal active center sites, and strong metal–support interactions. These unique advantages provide potential opportunities for improving catalyst efficiency and thermal stability [19]. Single-atom catalysts also have similar characteristics to homogeneous catalysts and heterogeneous catalysts, such as isolation of active sites in homogeneous catalysts and stable and easy separation of heterogeneous catalyst structures.

#### 2.2.1. High Selectivity

##### Metal Atom Alloying

A single-atom alloy (SAA) also has the advantages of high use of metal atoms [20], uniform distribution of active center sites, and unique geometric and electronic structure. It exhibits superior catalytic performance in heterogeneous catalytic reactions. Meng et al. [21] designed the Ru_1_Con alloy catalyst through a two-dimensional restriction strategy. The isolated ruthenium atoms are fixed on the surface of cobalt nanoparticles, and the catalyst has significantly improved activity and long-chain hydrocarbon selectivity in the FTS reaction at a lower reaction temperature. In addition, through in situ experiments, catalysis experiments, and density functional theory (DFT) calculations, the reaction mechanism is deeply explored. The results show that the higher electron density of single-atom Ru at the Fermi level and the synergistic effect with neighboring Co jointly reduce the rate-limiting barrier of the C–O cracking and C–C chain growth process, therefore having higher catalytic activity.

##### Atomic Interface Regulation Affecting Metal Atoms and Supports

Atomic interface regulation could affect the structure–property correlation based on the control of the coordination chemistry of individual atoms. A new strategy reported the use of direct atoms emitting from bulk metals, and subsequent trapping on nitrogen-rich porous carbon to form appropriate interactions between the metal atoms and supports. The central atom forms a coordination structure with other atoms, which changes the electronic structure of the central atom to increase its activity [22,23].

Wang et al. [24] prepared N and S co-coordinated Bi single-atom catalysts (Bi-SAs-NS/C) by anion and cation co-diffusion method, and the electrochemical catalytic performance of Bi-SAs-NS/C on CO_2_RR was studied. As the Bi source and S source, the Bi_2_S_3_ precursor can simultaneously diffuse into the surrounding polydopamine (PDA) layer. As the Bi source and S source, the Bi_2_S_3_ precursor can simultaneously diffuse into the surrounding polydopamine (PDA) layer. Both Bi and S are captured by the N site, and uniform N and S co-coordinated Bi active sites are formed at high temperature. The hollow structure provides sufficient channels for the diffusion and electron transfer of CO_2_RR related substances. The atomically dispersed Bi-N_3_S sites significantly improve the activity and selectivity of CO_2_RR. The experimental and theoretical calculation results show that the excellent catalytic performance of Bi-SAs-NS/C is due to the replacement of coordination N with low electronegativity S in the Bi-N_4_C center, which can reduce the activation energy of the reaction and promote the generation of rate-limiting step intermediates and accelerate the reaction rate.

#### 2.2.2. Less Consumption and Low Cost

A mono-atomic catalyst immobilized on the carrier has a higher degree of dispersion and a low load. Among the many carriers of SACs, carbon-based has excellent properties, such as low price, diverse structures, high stability, and good electrical conductivity. It is widely used as a carrier for SACs. There is a strong interface interaction between metal atoms and adjacent carbon atoms in the carbon skeleton, which will change the charge density and electronic structure of the metal atoms, and promote the formation of additional active sites by adjacent carbon atoms. Therefore, the carbon-based supported metal single-atom catalyst has an atomic use rate of nearly 100%. It has high efficiency in energy conversion. In addition, non-precious metal single-atom catalysts can reduce costs. By designing a unique catalyst structure, economic benefits can be created, and air and seawater can be converted into chemicals to achieve energy sustainability [25]. Xu et al. [26] developed a wastewater treatment system, using Cu(NO_3_)_2_ and dicyandiamide as precursors, through a simple one-pot synthesis of C_3_N_4_-supported copper single-atom catalyst (Cu-C_3_N_4_). The mono-atomic copper catalyst exhibits excellent catalytic activity in activating H_2_O_2_ to generate ·OH at pH = 7.0; an electrolysis device is designed, which can produce 10 g L^−1^ of H_2_O_2_. at a total cost of USD 4.66 per m^3^ by consuming air, electricity, and 0.1 M Na_2_SO_4_ electrolyte. The Cu single-atom catalyst can catalyze and activate H_2_O_2_ to generate hydroxyl radicals under the condition of pH = 7.0, without energy input, and shows strong stability in the filter device. The single-atom catalytic filter and the H_2_O_2_ electrolysis generator are connected in series to form a wastewater treatment system. The wastewater treatment system has low cost and high efficiency, and can reduce the energy and chemical requirements of advanced oxidation processes.

#### 2.2.3. Metal Atom–Carrier Interaction

SMSI has a significant impact on the activity, selectivity, and stability of the catalyst [27]. In the single-atom catalyst, the single-atom active center is more interacted by the metal support than the nano-catalyst. Han et al. [28] found that a Pt single atom can have a strong classical metal–support interaction with TiO_2_ support, but its required temperature is higher than that of Pt NPs. Coordination saturation causes Pt single atom to lose CO adsorption capacity. The strong metal–support interaction can change the morphology and electronic structure of metal atoms, so that the catalyst has unique properties. SMSI can cause charge transfer and mass transfer between the carrier and the metal. SMSI helps to explore the reaction mechanism and synthesis of single-atom catalysts with superior performance.

#### 2.2.4. Higher Surface Free Energy

Single-atom catalysts have high chemical activity, but due to the small surface area and high surface free energy, the catalyst tends to reduce its own energy. It is easy to migrate, aggregates to form large clusters, and leads to inactivation, and the structure is fragile. Based on first-principles calculations, Yang et al. [29] designed a stable and efficient transition metal (TM) single-atom catalyst protected by the two-dimensional material graphene or carbon nitride as the “armor”. A single TM atom is closely attached between graphene (GR) layers or GR-C_3_N layers to form a stable sandwich structure. Studies have found that the binding energy and migration energy barrier of GR-TM-GR and C_3_N-TM-GR are much higher than those of direct adsorption type single-atom catalysts, indicating that the “armor” single-atom catalysts are highly stable in terms of thermodynamics and kinetics. A single atom can transfer charge to the C or N atoms in the “armor” layer on the surface near it, transfer its high catalytic activity to the armor, and promote chemical reactions on the surface. A single atom can transfer charge to the C or N atoms in the “armor” layer on the surface near it, transfer its high catalytic activity to the armor, and promote chemical reactions on the surface. The “armor” type single-atom catalyst can also efficiently catalyze the hydrogen evolution reaction, and the free energy barrier of the reaction is almost 0 (~0.01 eV), which is better than commercial Pt catalysts (~0.09 eV). It effectively integrates the advantages of the stability of two-dimensional materials and the high reactivity of single-atom catalysts.

Zhou et al. [30,31] realized the preparation of metal single-atom catalysts by adjusting the free energy of the oxide surface, and investigated its reaction performance. The surface free energy of bulk metals is much higher than that of oxides, causing metal atoms to easily agglomerate on the surface of oxides. When the oxide is thinned to a single layer, its surface free energy will be adjusted by the base metal to approach the metal surface free energy, and the metal deposited on the thin layer of oxide will be in a highly dispersed state. A single layer of CuO film was grown on the surface of Cu(110) single crystal, and platinum (Pt) metal atoms were deposited to obtain a thermally stable Pt single-atom model catalyst above 400 K. The preparation method of the supported metal single-atom catalytic system is simple and easy to implement, and does not need to stabilize the metal single atom by adsorbing molecules or intercalating into the crystal lattice.

Although single-atom catalysts have many advantages, they also have some limitations. When the size of the nanoparticle is reduced to the single-atom level, the free energy of the metal surface is greatly increased. During the preparation and reaction process, the single atoms are easy to aggregate and couple to form clusters, which leads to catalyst deactivation and affects the activity and stability of the catalyst. In addition, single-atom catalysts have high requirements for structural characterization instruments and complex analysis of catalytic reaction mechanism, which is an engineering technical problem. It is also difficult to maintain the high stability and high activity of single-atom sites in the design of SACs. In some reactions, the strong interaction between the intermediate product and the transition metal single atom may deactivate the reaction site. In the process of applying defect engineering to synthesize single-atom catalysts, conjugation will be weakened, leading to electron-hole recombination, and will also damage the charge transport and destroy the carrier structure, affecting its performance.

## 3. Preparation of a Single-Atom Catalyst

### 3.1. Atomic Layer Deposition Method

Atomic layer deposition, abbreviated as ALD, is a method in which the carrier material is alternately exposed to pulsed vapor of different reaction precursors, and the substance is deposited on the surface of the carrier atomically in a self-limiting manner. ALD was originally used for the preparation of polycrystalline fluorescent materials ZnS:Mn and amorphous Al_2_O_3_ insulating films, and has now been successfully used for the preparation of metal single-atom catalysts on the surface of supports such as graphene.

The advantages of ALD are that the deposition parameters are precisely controllable, the deposition uniformity and repeatability are good, and it can precisely control the growth of different materials in an atomic layer to form composites with different morphologies, providing an ideal model catalyst for basic catalysis research, and then explore the effect of particle size, carrier surface characteristics, and coating on the surface of metal or alloy nanoparticles on catalytic performance. ALD is an important means to study the synthesis of supported single-atom catalysts and the structure–activity relationship. The disadvantage is that the prepared single-atom catalyst has low stability and high cost, and this technology is not suitable for commercial large-scale preparation.

Li et al. [32] has prepared Pt_1_ SACs by atomic layer deposition (ALD) which relies on continuous molecular-level self-limiting surface reaction technology, and the 5*d* state of Pt_1_ single atom on Co_3_O_4_ was adjusted through strong electronic metal–carrier interaction (EMSIs). At room temperature, in the ammonia borane dehydrogenation reaction, the activity of the catalyst is significantly higher than that of other supports. The catalyst is different from the Pt nanoparticle catalyst in that it does not deactivate quickly and has good stability and anti-sintering properties. Spectral characterization results show that EMSI regulates the adsorption of ammonia borane and promotes the hydrogen desorption reaction by adjusting the unoccupied 5d state of Pt_1_ single atom. Therefore, Pt_1_ SACs exhibit high activity. In the Pd_1_/Co_3_O_4_ hydrogenation reactions, SMSI occupies an important position and exhibits a unique electron-promoting effect.

Song et al. [33] proposed an atomic layer deposition (ALD) strategy to synthesize Pt SACs on metal–organic framework (MOF)-derived N-doped carbon (NC). By adjusting the ALD time of the Pt precursor, a Pt single-atom catalyst was prepared on the MOF-NC support. X-ray absorption fine structure spectroscopy determined the electronic environment in Pt SACs, and it was found that electron vacancies increased, forming a Pt-N coordination structure. Under the synergistic effect of the low coordination environment of the MOF-NC carrier and the interaction between the platinum atom and the nitrogen doping site, Pt SACs have higher activity and stability in promoting the oxygen reduction reaction (ORR). Density functional theory calculations show that platinum single atoms are more likely to be fixed by carbon sites doped with pyridine nitrogen. The adsorbed hydroxyl and oxygen jointly regulate the electronic structure of Pt-SACs, so that the free energy change of the rate-determining step is drastically reduced.

It is important to use a precursor to settle a single atom without oxygen and then to burn off the precursor with ozone. The exposure time is another parameter to control in the ALD method that will affect the size, controlled from single atoms to subclusters and nanoparticles.

### 3.2. Co-Precipitation Method

The co-precipitation method refers to the preparation method of adding a precipitating agent to a solution containing two or more cations, and obtaining a supported metal catalyst in which different active species are uniformly dispersed through a precipitation reaction. It is relatively mature and widely used in the preparation of metal single-atom catalysts and nano-scale metal catalysts. The advantage of this method is that since the precursors in the solution are uniformly mixed, the co-precipitation method is an important method for preparing a composite oxide that contains two or more metal elements and is uniformly dispersed.

The disadvantage of this method is that the metal loading is low; multiple parameters in the catalyst preparation process have a great influence on the performance of the catalyst and need to be strictly controlled. These influencing factors include the addition rate of the precursor, the size of the droplets, the degree of stirring, and the reaction temperature, pH, and reaction time, etc. In addition, some metal atoms will be buried in the carrier aggregation interface or the bulk carrier. These buried single metal atoms cannot contact the reaction molecules and participate in the reaction, which greatly reduces the overall performance of the single-atom catalyst.

Qiao et al. [34] prepared a new type of Au_1_/FeOx single-atom catalyst by co-precipitation method, and FeOx support was synthesized by co-precipitation method. In the CO oxidation reaction, the catalyst exhibits excellent performance with high activity and strong stability. It also has ultrahigh stability at 400 °C. The experimental and theoretical results show that the strong covalent metal–support interaction (CMSIs) between a single Au_1_ atom and FeO_x_ lattice oxygen atoms is due to the positive charge of the single Au_1_ atom located in the Fe vacancy of FeO_x_ nanocrystals, so the Au_1_ atom became stably fixed on the carrier.

Max et al. [35] synthesized precipitated heteropoly acid by co-precipitation method. The atom-dispersed Rh is immobilized on the phosphotungstic acid carrier. In the CO oxidation process, the identification of several key intermediates and the steady-state structure of the catalyst indicate that the reaction follows the unconventional Mars–van Krevelen mechanism, and the activation of O_2_ is rate-limited. In situ XPS confirmed the contribution of the heteropoly acid carrier, while in situ DRIFT spectra proved the oxidation state of Rh and CO adsorption. By observing the metal center and the key components of the carrier, we can clearly understand the oxidation process of CO on the atom-dispersed Rh sites.

### 3.3. Dipping Method

The impregnation method is one of the classic preparation methods of traditional heterogeneous catalysts: after the support is in contact with the aqueous solution of the metal salt, the metal salt solution is adsorbed or stored on the surface of the carrier or in the pore structure, the excess solution is removed, and then the catalyst is prepared by drying, calcination, and activation. The impregnation method is not suitable for preparing mono-atomic metal catalysts with high metal loading, but it is suitable for preparing mono-atomic metal catalysts on open supports, especially separate nanostructures. The low-volume impregnation method is simple to operate and does not waste expensive metal raw materials, but it is difficult to ensure the uniform dispersion of individual metal atoms on the surface of the carrier. The excessive impregnation method depends on the adsorption capacity of the precursor complex on the surface of the carrier. Therefore, the precursor–carrier interaction is very critical, and the metal loading and dispersibility are strongly related to the characteristics of the anchor points on the surface of the carrier. When the oxide carrier is dispersed in an aqueous solution, it will be polarized and charged, which is controlled by the pH of the solution. In an acidic solution, the surface adsorption sites (M-OH) are positively charged and covered by anions. In alkaline solutions, the surface adsorption sites (M-OH) are negatively charged and covered by cations. According to the Brunelle adsorption model, the key parameters that control the dispersibility of metals include: (1) the type and concentration of the metal precursor; (2) the pH of the aqueous solution; (3) the type of support and its surface functional groups. Coordination ions of various metal precursors and non-aqueous solvent immersion solutions can control specific adsorption behavior. [M-Cl] coordination ions can form hydrates and hydrolyze to achieve metal adsorption.

Rassolov et al. [36] synthesized the mono-atomic alloy Pd-Ag catalyst by wet impregnation method, and studied the catalytic performance of the Pd_1_Ag_3_/Al_2_O_3_ mono-atomic alloy catalyst, a kind of SAA catalyst, in the liquid phase hydrogenation of diphenylacetylene. The results show that the Pd_1_Ag_3_/Al_2_O_3_ has good selectivity to stilbene. It remains stable within the larger conversion rate range of DPA. The selectivity of Pd_1_Ag_3_/Al_2_O_3_ to stilbene decreased steadily with the increase of DPA conversion rate.

Su et al. [37] loaded platinum and cobalt on HZSM-5 by impregnation method to synthesize Pt-Co/HZSM-5 three-way catalyst for the catalytic oxidation of dichloromethane (DCM). The CO_2_ yield of the catalyst is higher, and the yield of by-products is lower. The activity remains stable without significant changes. The characterization results show that Pt atoms can be stably fixed on Co (in the form of Co_3_O_4_) to achieve single-atom dispersion of Pt, which can increase the proportion of oxygen vacancies on the surface of Co_3_O_4_ and enhance the redox performance of Co_3_O_4_. The research results show that the dissociation and deep oxidation of DCM is promoted by the synergy between Pt, Co, and HZSM-5.

### 3.4. Low-Temperature Photoreduction Method

The photochemical reduction method uses ultraviolet light to replace the electrochemical reduction process. The mechanism uses high-energy ultraviolet light to irradiate the reaction solution to generate free radicals with strong reducing ability to carry out the reduction reaction. This method has the advantages of simple synthesis, low cost, and low toxicity. However, the particle size uniformity is not good. The photochemical reduction method requires that the system must contain substances that can absorb ultraviolet light and release electrons. This method also has some common disadvantages including: (1) the absorption range of ultraviolet light is narrow, the use rate of light energy is low, and the efficiency is also limited by the catalyst, wavelength, and reaction vessel; (2) the electron-hole pairs generated by light are easy to recombine and cause the catalyst to deactivate.

Guo et al. [38] used ceramic MOF (Ce-MOF) with specific defects in the nanoporous structure and crystal structure as the carrier; the low-temperature photoreduction method is used to prepare separated Pt atoms, then it is preferentially located in the defective metal oxide unit lacking organic linkers in Ce-MOF. The Pt single-atom modified Ce-MOF catalyst was obtained. The characterization results show that isolated noble metal atoms can be strongly coupled with the cerium oxide unit in the porous Ce-MOF, providing abundant reaction sites in the entire porous crystal structure. SACs can achieve high activity, rapid reaction kinetics, and thermal stability, and only consume 10% of the precious metal required by the precious metal catalyst. There is no high temperature and high-energy process in the synthesis process, and these catalysts can be prepared on a large scale. Ce-MOF-mediated Pt atom catalyst was used for CO oxidation. Due to the uniform dispersion and unique electronic hybridization with Ce-MOF, the low-content conjugated catalyst exhibits a 100% CO conversion rate at a low temperature of 150 °C. In addition, the catalyst is still active even at a low temperature of 40 °C, and can withstand temperatures up to 300 °C for a long time, with a conversion rate of 100%. Studies have shown that the porosity and crystallinity of MOF provide strong interactions and rapid reaction kinetics for gaseous reactants and products. The strong coordination between the noble metal and the cerium oxide unit in Ce-MOF not only improves the stability of the catalyst, but also controls the activity of the separated catalyst, therefore increasing the atom use of the noble metal.

### 3.5. Flame-Spray Pyrolysis Method

Flame-spray pyrolysis is a gas-phase synthesis method. The metal salt solution is atomized by an atomizer and sprayed into a high-temperature flame reactor formed by hydrogen or alkanes.

The solvent evaporates and, when accompanied by the thermal decomposition of the metal salt, the solution is in a supersaturated state, and the different substances in the solution are combined to precipitate solids to obtain fine nanoparticles, which can be used to synthesize the catalyst in one step. The flame temperature determines the metal loading of the catalyst. This method is a scalable technology that can precisely control the composition, crystallinity, and particle size of the nanoparticles. The prepared catalytic material has a higher specific surface area and evenly dispersed metals, which significantly enhances the catalyst activity [39].

Ding et al. [40] used high-temperature process conditions (>1000 °C) to prepare thermally stable metal oxide supported mono-atomic Pt catalysts by flame-spray pyrolysis. ZrO_2_ is the best carrier for stable atom dispersion of Pt. Compared with the mono-atomic Pt catalyst synthesized by the traditional impregnation method, the catalyst prepared by this method exhibits excellent catalytic performance in CO oxidation, methane combustion, and methane partial oxidation reactions. The characterization results show that the preparation method can make ZrO_2_ form a tetragonal monoclinic crystal phase structure, improve the oxidation-reduction ability, and thus significantly enhance the high-temperature catalytic activity.

### 3.6. High-Temperature Cracking Method

The synthesis temperature of conventional single-atom catalysts is generally low, and there is no strong bonding between the single atoms and the support, so agglomeration is prone to occur during use (high temperature or high voltage), resulting in catalyst deactivation [41]. The high-temperature cracking method is currently the main method for preparing non-precious metal-supported catalysts. High-temperature treatment can promote the formation of a covalent bond between the metal atoms and the carrier, can strongly promote the solidification of the metal matrix and make the single atoms have a higher degree of dispersion, which greatly enhances the stability of the single atoms.

Huang et al. [42] reported a high-temperature self-assembly method to prepare thermally stable silver (Ag) single-atom catalysts. The single-atom site of silver is fixed on antimony-doped tin oxide (ATO) by atom-trapping in the air at 800 °C. After high temperature, Ag has a unique self-dispersion on the ATO carrier, which is opposite to the tendency of sintering or coalescence. The study also confirmed that there are a large number of scattered isolated Ag species in the samples calcined at Ag/ATO-800 °C. CO oxidation experiments show that after calcination at 800 °C, the single-atom Ag-on-ATO catalyst is still very stable, with negligible decay and increased activity because the single silver atom is tightly bound to the ATO carrier, and the reaction occurs at the binding site. This discovery paves the way for the use of commercial carriers to disperse and stabilize precious metal single atoms through atom trapping in the CO oxidation reaction, and maximize the metal use efficiency in the CO oxidation reaction.

Han et al. [43] prepared a single-atom catalyst with high nickel loading (5.92 wt.%) by polymer-assisted pyrolysis. The synthesis method constructs an ultrathin nanosheet structure, and adds a fluorine dopant to adjust the electronic structure (Ni-N_4_) of the metalloporphyrin-like active site. Studies have shown that this unique structure of the catalyst has significant electrocatalytic performance for the conversion of CO_2_ to CO. The results of ATR-IR and theoretical calculations also show that doped fluorine can adjust the electronic configuration of the central Ni-N_4_ site, therefore reducing the CO_2_ activation energy barrier, which is conducive to the formation of key *COOH intermediates. This research provides important reference value for the design and synthesis of single-atom catalysts with high activity, strong selectivity, and excellent durability.

Zhao et al. [44] proposed a simple strategy for the synthesis of mono-atomic iron catalysts. This method embeds iron catalysts in porous carbon nanomaterials doped with nitrogen and fluorine, with Fe-N_5_ active sites. Iron porphyrin (DFTPP) FeCl (DFTPP = 2,6-difluorotetraphenyl porphyrin) and imidazole are coordinated and assembled on carbon black with polyvinylimidazole (PVI) coating, and the carbon black is heat-treated to obtain a catalyst (namely C@PVI-(DFTPP)Fe-800). The ORR performance of the catalyst is significantly enhanced. The fluorine atoms doped in the carbon support can adjust the ORR activity of the catalyst. This method opens up a new way for the preparation of structure- and performance-related electrocatalysts with many Fe-N_x_ active sites at the molecular level.

### 3.7. Underpotential Deposition Method

The underpotential deposition method is a surface-limited redox replacement reaction, which refers to the phenomenon that a metal is deposited on another substrate under the condition of a positive thermodynamically reversible potential, forming a deposition layer of atomic thickness [45]. The underpotential deposition (UPD) method can controllably synthesize metal nanocrystals and improve their catalytic performance, and can also modify the surface of metal electrodes at the atomic scale [46].

Xu et al. [47] successfully attached mono-atomic copper to the sulfur sites of the doped graphite foam through an underpotential deposition strategy. This synthesis strategy is a simple and scalable electrochemical strategy. The loading of Cu SACs can be adjusted by changing the deposition potential, the content of S in the doped graphite foam and the concentration of the Cu precursor; in addition, the law conforms to the Nernst equation and Langmuir adsorption model, and developed stripping voltammetry to quantitatively analyze the synthesized Cu SACs. The catalyst has high oxygen reduction reaction (ORR) activity and long-term stability.

Shi et al. [48] reported surface-limited room-temperature electrodeposition technology. Using a specific substrate positioning strategy, the rapid preparation of atomic-level dispersed metal catalysts at room temperature was achieved. This strategy can be extended to at least 12 metal atom systems. Taking the stripped MoS_2_ as the substrate, the non-noble metal single atoms are grown by underpotential deposition and anchored by sulfur atoms to achieve its thermodynamic stability. Then, through the current replacement strategy, the noble metal single atom is replaced, to obtain the target atomic-level dispersed metal catalyst. The fixed-point electrodeposition can form a thermodynamically favorable metal–support bond, and then the continuous formation of the metal bond is automatically terminated, therefore avoiding the aggregation and growth of metal single atoms. The prepared Pt-SAs/MoS_2_ atomic-level dispersed metal catalyst showed excellent activity and stability in the HER reaction.

### 3.8. Ball-Milling Method

The ball-milling method grinds the particles continuously for a long time, and the particles are pulverized by strong collisions, and the shape and size of the particles are changed. The mechanochemical ball-milling method can make the catalyst have a series of unique properties. The mechanical collision makes the catalyst produce high defect density in situ, which can improve its activity and accelerate the progress of the chemical reaction [49].

Gan et al. [50] used two acetylacetonates with very similar properties as raw materials. A mono-atomic alloy catalyst with platinum distributed on the atomic level of cobalt nanoparticles (i.e., Pt_1_/Co) was successfully realized by ball-milling. The catalyst exhibits excellent catalytic performance for the hydrodeoxygenation of methyl furfural to DMF. After reacting at 1.0 MPa H_2_, 180 °C for 2 h, the conversion rate of hydroxymethyl furfural was 100%, and the selectivity to DMF was 92.9%. The study of the reaction pathway shows that the hydrogenolysis of the C=O bond in HMF to form 2,5-dimethylolfuran is the main method for the hydrodeoxygenation process, and it is promoted by the synergistic effect of Pt and Co. After running continuously for five times, it showed excellent stability without agglomeration. The method realizes the mass production of kilogram-level Pt_1_/Co, is simple and efficient, has no solvent, and has no size amplification effect, so that it has practical application potential.

He et al. [51] mixed a certain proportion of palladium acetylacetonate and zinc acetylacetonate (1:400) by ball-milling, and then calcined at high temperature to obtain Pd_1_/ZnO palladium single-atom catalyst. Spherical aberration electron microscopy and synchrotron radiation technology show that Pd is distributed on ZnO in the form of atomic dispersion. The study found that for the two probe reactions of alkyne semi-hydrogenation and carbon monoxide oxidation, there is basically no difference in activity, selectivity and stability of single-atom catalysts of different preparation levels, and the catalytic performance remains unchanged. The single-atom catalyst synthesized by the method has no amplification effect and is suitable for large-scale batch synthesis. By investigating the structure of the catalyst obtained by different ball-milling time, it was found that with the extension of ball-milling time, a transitional change of the aggregation state of palladium species from nanoparticle to nanocluster to atomic-level dispersion was observed. It shows that the ball-milling process promotes the high dispersion of palladium.

### 3.9. Molten Salt Method

The melting method fuses the reactants to form a mixture under high-temperature conditions. The advantage of this method is that the different components are fully mixed and evenly distributed, high temperature can promote the diffusion between the components, make them fully contact and complete the reaction, and the synthesized catalyst has a smaller particle size.

Cao et al. [52] used the molten salt-assisted thermal emission and trapping method. Massive iron oxide (III) powder is converted into an efficient single-atom iron catalyst for the cathode oxygen reduction reaction. Anions and cations have strong polarity, which can effectively promote the breaking of bulk Fe_2_O_3_ chemical bonds and the volatilization of Fe. Nitrogen-doped porous carbon fixes the vaporized Fe molecules on the surface to form a mono-atomic dispersed “Fe-N_4_-O_2_” site catalyst. The catalyst has stronger activity and better stability in oxygen reduction reaction in alkaline media; it has superior performance in a zinc–air battery. The study also found that this method can also be used to prepare non-noble metal single-atom catalysts (metal = Co, Mn, Cu, Ni) by changing the precursors of different metal oxides.

Xiao et al. [53] designed a new molten salt method for the synthesis of single-atom co-catalysts. By supporting single-atom Ni co-catalysts on TiO_2_ nanoparticles, the photocatalytic performance of TiO_2_ was significantly enhanced. The liquid environment and space restriction provided by molten salt promote the uniform dispersion of Ni ions on the surface of TiO_2_, while the strong polarization provided by molten salt promotes the formation of strong Ni-O bonds between Ni ions and O atoms on the surface of TiO_2_ particles, therefore avoiding the agglomeration of Ni atoms on the surface of TiO_2_ particles. In addition, Ni atoms help to form oxygen vacancies on the surface of TiO_2_ particles. The synergistic effect of the mono-atomic Ni promoter prepared by the method and the oxygen vacancies makes the Ni/TiO_2_ photocatalyst exhibit efficient and stable photocatalytic hydrogen production performance. In addition, the steric confinement effect of molten salt leads to a high degree of dispersion of TiO_2_ nanoparticles. At the same time, due to the strong polarization provided by the molten salt, the surface of the nano-TiO_2_ becomes metastable. Salt (LiCl and KCl) can be reused after being recovered from water, which makes this method green and low cost, suitable for large-scale production.

## 4. Characterization Technology of Single-Atom Catalyst

The morphology of the catalyst is the focus of its research. The performance of the catalyst is not only related to its particle size, but also inseparable from its morphology. Its composition, structure, particle size, and morphology and other microstructures largely determine its macroscopic properties [54]. The characterization of the catalyst helps to understand its composition, structure, and physicochemical properties.

### 4.1. Electron Microscopy Characterization

Electron microscopy technology determines particle size, particle size distribution, and dispersibility of the catalyst by observing the fine morphological structure on the surface of the catalyst. Commonly used electron microscope characterization methods mainly include scanning electron microscope (SEM), high-resolution transmission electron microscope (HRTEM), atomic force microscope (AFM) [55], scanning tunneling microscope (STM), etc.

Wang et al. [56] formed Cu single atom on UiO-66-NH_2_ carrier by light induction method, and synthesized Cu SAs/UiO-66-NH_2_. The photocatalytic CO_2_ reduction performance is significantly enhanced, and it can be efficiently reduced to liquid fuel. In addition, to characterize it, the TEM image of the Cu SAs/UiO-66-NH_2_ sample shows that there are no Cu nanoparticles on the surface of the carrier. Through aberration-corrected high-angle circular dark-field scanning TEM (ACSTEM) measurement, the morphology of Cu on the carrier is determined. It shows the atom Cu is dispersed on the carrier. By element map analysis, the Cu is distributed on the UiO-66-NH_2_ carrier with a high degree dispersion. The Cu SAs/UiO-66-NH_2_ element map shows that Cu is evenly distributed on the UiO-66-NH_2_ carrier and has a high degree of dispersion.

Xu et al. [26] prepared a Cu-C_3_N_4_ single-atom catalyst by a simple one-pot method, and the morphology of the Cu-C_3_N_4_ sample was determined by HR-TEM. The characterization results showed that the catalyst exhibited an amorphous structure, uniformly distributed, Cu or CuO nanoparticles were not observed. The aberration-corrected high-angle dark-field scanning transmission electron microscope (HAADF-STEM) was used to more accurately characterize the catalyst material, it was found that only single copper atoms were distributed on the carrier in isolation, and no copper nanoparticles existed.

### 4.2. Spectroscopy Analysis

Spectroscopy analysis mainly includes infrared spectroscopy (FTIR), Raman spectroscopy (Ram), ultraviolet-visible absorption spectroscopy, fluorescence spectroscopy, nuclear magnetic resonance spectroscopy, X-ray photoelectron spectroscopy X-ray diffraction [57], Fourier transform infrared spectroscopy (FTIR) [58], diffuse reflection infrared Fourier transform spectroscopy (FLEDS), electrochemical impedance spectroscopy (EIS) [59], X-ray absorption near-edge structure (XANES) spectrum, energy dispersive X-spectrum [60], etc. Characterization of the catalyst by spectroscopy technology can explore the physical and chemical properties of single-atom catalysts, also reveal the significant differences between single-atom catalysts and traditional nano-metal catalysts, and further clarify the catalytic reaction mechanism.

Gan et al. [61] polymerized dopamine hydrochloride with an iron template to synthesize an ultrathin single-atom bifunctional Fe-Nx-C catalyst, and characterized it. The existence state of Fe atom in Fe-N_x_-C-SACs was preliminarily analyzed by inductively coupled plasma atomic emission spectroscopy (ICP-AES), X-ray photoelectron spectroscopy (XPS), X-ray diffractometry (XRD), and Raman spectroscopy. The results show that Fe atoms are isolated and dispersed, stably attached to the vacancies of the nitrogen-doped carbon matrix, forming abundant Fe-N_4_ sites. The diffraction peaks of Fe or FeO_x_ species are not observed in the XRD spectrum of Fe-N_x_-C-SACs; only the characteristic bands of the carbon matrix appear in the Raman spectrum, indicating that there is no crystalline iron species.

X-ray photoelectron spectroscopy (XPS) mainly provides information about the valence state of the catalyst and its chemical environment. Jing et al. [62] used inorganic sulfur anion-assisted SMS method to synthesize cobalt-based C-SACs with different coordination structures. Using high-resolution X-ray photoelectron spectroscopy (HR-XPS) to conduct in-depth research on Na_2_SO_4_ and Na_2_S_2_O_3_-based samples, it was found that the surface composition and chemical structure of the two samples are very different. The existence of C-N/C-S bond (285.6 eV) in the C 1s spectrum indicates that the two samples were successfully co-doped with N/S. The N 1s spectra show that pyridine N and pyrrole N play an important role in the two samples. In the S 2p spectrum, the sulfur oxide and thiophene-S peaks of S1-N-Co_1_/C and S2-N-Co_1_/C from the s 2p_3/2_ and s 2p_1/2_ multiplexes can be observed. The C-S-C signal can be observed, indicating that S is successfully doped in the carbon lattice. The Co-S bond was detected in S1-N-Co_1_/C, indicating that the S and Co in S1-N-Co_1_/C are tightly bound, and the two samples have different coordination structures.

## 5. Application of Single-Atom Catalysts in the Environmental Field

### 5.1. Applications in the Field of Electrochemistry

#### 5.1.1. CO Oxidation

When metals are dispersed in the form of single atoms, they are more active in the CO oxidation reaction, and are more active at low temperatures, and single-atom sites are used as the active sites for the CO reaction.

Many theoretical and experimental studies have been carried out on single-atom catalysts for CO oxidation, including noble metal [63,64] and non-noble metal [65,66,67,68,69] single-atom catalysts. There is a strong metal–carrier interaction between the metal atom and the carrier. The unique structure of the single-atom catalyst promotes the activation of the adsorbed CO on the surface, and at the same time inhibits the occurrence of side reactions. There is an electron-transfer effect between the metal atom and the carrier, which plays an important role in the adjustment of the electronic structure of the catalyst, therefore enhancing the catalytic activity and improving the selectivity of the CO oxidation reaction.

Sarma et al. [70] studied the role of a series of uniformly structured transition metal single-atom catalysts attached to the MgO support in the catalytic reaction of CO oxidation, indicating that the metals in the single-atom catalysts have the characteristics of specific reactions.

Cui et al. [71] studied CO/O_2_ adsorption and CO oxidation on Pd-doped C_3_N (Pd-C_3_N) monolayers, and a low-temperature and high-activity catalyst for CO oxidation was proposed. The Pd dopant is stably fixed on the N vacancy of the C_3_N monolayer, forming a large binding energy, and there is no possibility of clusters. The adsorption energy of Pd-C_3_N monolayer for O_2_ is greater than that of CO molecules, and the Eley–Rideal (ER) mechanism is the first choice for CO oxidation. The Langmuir–Hinshelwood (LH) mechanism is also used to fully understand the CO oxidation process. At room temperature, the Pd-C_3_N monolayer also has good versatility in the oxidation of CO, which provides a new idea for the development of a new low-temperature and high-efficiency single-atom catalyst (SAC) based on the C_3_N monolayer.

#### 5.1.2. Oxygen Reduction Reaction

In various sustainable energy storage and conversion devices, oxygen reduction reactions occur as a key step in determining energy conversion efficiency [72]. An important reaction in fuel cells and zinc–air batteries is the oxygen reduction reaction. Efficient noble metal-free oxygen reduction reaction electrocatalyst is the key to reducing the cost of fuel cells and metal–air batteries [73]. It is found that Pt-based catalysts show good catalytic performance in an oxygen reduction reaction. Pt is usually used as a typical electrocatalyst for ORR [74]. Fixing a single atom of Pt on the carrier can minimize its required amount, but the adsorption capacity for O_2_ is reduced and the catalytic activity is degraded [75]. Pt can be dispersed on α-Fe_2_O_3_ to prepare a single-atom catalyst with a pair of active centers, which has high activity and stability in the ORR reaction. To improve the activity of single-atom electrocatalysts in the oxygen reduction reaction, considerable research has been carried out, such as adjusting the coordination structure of the metal single-atom active center, increasing the central metal loading, and changing the electronic structure and porosity of the carrier. Noble metal catalysts can promote the slow four-electron-transfer ORR process, but their large-scale use increases the cost of fuel cells. At present, to reduce the cost of ORR reaction, it has become a trend to use non-noble metal electrocatalysts instead of precious metal catalysts. There is bond cooperation between metal atoms and C, N, and O atoms on carbon or dopant-based carbon supports, which can be used to synthesize highly dispersed single-atom catalysts with metal active centers, thus achieving the goal of improving the use rate of metal precursors and enhancing the activity of central metal atoms.

Han et al. [76] developed a template casting method using KIT-6, Fe(II)-Phen, and 2-Mi as precursors to prepare a highly efficient single-atom Fe-N-C/N-OMC electrocatalyst with Fe-N-C sites embedded in a three-dimensional (3D) N-doped ordered mesoporous carbon framework. The ORR of Fe-N-C/N-OMC in alkaline electrolyte showed 0.93 V higher than Pt/C for half-wave potential, 0.85 V (A cm^−2^) for kinetic current density, 38.4 s^−1^ for turnover frequency and 66.4 A mg_Fe_^−1^ for mass activity. DFT calculations showed that the graphite N dopant changed the 3D state of Fe, provided better adsorption energy for oxygen-containing intermediates of the FeN_4_ active site, and improved the intrinsic ORR activity of FeN_4_ site. The high Fe and N loading, the three-dimensional ordered mesoporous structure of carbon skeleton and the high electrochemically active surface area make the Fe-N-C/N-OMC have dense active centers. As a positive electrode for zinc–air batteries, Fe-N-C/N-OMC has high voltage, high power density, and excellent durability. It shows excellent performance in oxygen reduction reaction and zinc–air battery.

Zhou et al. [77] designed and prepared a novel three-dimensional Co single-atom catalyst (CO_SA_-N-C) with unique star structure and Co-N_3_C coordination structure. The catalyst has six equibranched configurations of the hybrid MOF precursor. The active sites are abundant and can be directly connected to the reactants in the electrolyte. The hybrid MOF precursors have a special metal-to-ligand ratio, with the active Co-N_3_C scattered on the N-doped carbon. The catalyst exhibited good ORR catalytic activity and electrochemical stability in oxygen reduction and zinc–air batteries. DFT calculations also show that the hybridization of new electronic states with oxygen molecules is caused by the unique coordination in the catalyst. The catalyst performs well as a cathode in zinc–air batteries and has great potential for energy storage and conversion.

#### 5.1.3. Catalytic Oxidation of NO_x_

NO_x_ is a major atmospheric pollutant, which can cause environmental problems such as photochemical smog and acid rain. The removal efficiency of NO_x_ by traditional catalytic reduction method is low at low temperature. Therefore, it is very important to construct a novel and efficient catalyst for the catalytic oxidation of NO. Studies have found that adding NH_3_ adsorbent to the NO catalytic oxidation system can significantly increase the NO oxidation rate. However, due to the high cost of NH_3_ reduction, the development of economical and green catalysts is the key. Through first-principles calculations, Li et al. [78] found that the Cr single-atom catalyst supported on graphene is the most promising single-atom catalyst for NO oxidation. Compared with precious metals, the 3D orbitals of transition metals Fe and Co do not form paired electrons, so they have unique activities [79]. In addition, the synergy between the double active sites of metal atoms can also be used to catalyze the oxidation of NO, so that the adsorption of NO and O_2_ on the active surface is enhanced, and at the same time, the occurrence of side reactions is inhibited. The introduction of dual-active-site single-atom catalysts can effectively alleviate NO emissions, which is of great significance in the field of environmental catalysis.

Tian et al. [80] discussed the stability of the mono-atomic manganese catalyst supported by graphene nanomaterials and the adsorption behavior and catalytic activity of NO_2_, NO, and NH_3_. In addition, the results of adsorption energy and bond length show that the catalytic oxidation of NO_x_ by Mn/GS is realized by activating the N-O bond of NO_2_, and the main role of Mn/GS is to promote the adsorption of NO and NH_3_. The electron-transfer mechanism of NO_2_, NO, and NH_3_ adsorbed on Mn/GS was also studied. In the adsorption system, NO_2_ and NO act as electron acceptors, while NH_3_ acts as an electron donor. Finally, DOS and PDOS calculations were performed to confirm the formation of a chemical bond between N and Mn atoms.

#### 5.1.4. Water-Splitting

As a simple and efficient method for hydrogen production and oxygen production, electrocatalytic water-splitting is widely used in industrial production [81]. The use of catalyst electrocatalytic water-splitting is an effective way to prepare hydrogen fuel [82], which can realize the conversion and use of high-efficiency renewable energy [83]. The catalytic efficiency of the catalyst is of great significance in the study of electrocatalytic water-splitting. The development of efficient and durable electrocatalysts to accelerate the slow kinetic water-splitting reaction is an urgent problem to be solved. The precious metal Pt has strong activity in the electrocatalytic hydrogen evolution reaction, but due to its high cost and scarcity, its application in electrocatalysis has been severely hindered [84]. Water-splitting consists of two half-reactions of OER and HER. The preparation of single-atom catalysts with multifunctional active sites is the focus of current research. The synergy of multiple sites can simultaneously promote high-efficiency OER and HER, and improve the overall water-splitting capacity [85].

Yang et al. [86] studied the absorption characteristics of water molecules and the water-splitting reaction pathway of four kinds of graphene matrix-supported single-atom Fe catalysts. The single-atom catalyst exhibits different electronic properties on different substrates. The results show that the strong chemical adsorption of Fe/GS on water molecules is due to the high hybridization and overlap of the 3D orbitals of Fe single atoms and the 2p orbitals of O atoms, and the strong metal–support interaction plays an important role in the water-splitting reaction.

Zang et al. [87] reported a surfactant template-assisted method for the preparation of nickel mono-atomic electrocatalysts (Ni-SA/NC) with dual coordination structures (Ni-N_3_-O_2_ and Ni-N_4_). The electrocatalytic seawater cracking under alkaline pure water and seawater conditions was studied systematically. Based on X-ray absorption fine structure (XAFS) analysis and density functional theory (DFT) calculations, the authors found that Ni-N_3_-O_2_ is reduced to Ni-N_3_ before HER-catalyzed reaction. Compared with Ni-N_4_ coordination structure, this structure is more conducive to water dissociation and hydrogen adsorption. This work provides a theoretical basis and research direction for the application of a single-atomic catalyst in seawater electrolysis, and opens up a new design of new seawater electrolysis and a large-scale electrocatalyst for hydrogen production.

#### 5.1.5. CO_2_ Reduction Reaction

The single-atom catalyst electrocatalytic CO_2_ reduction reaction can convert greenhouse gases into high value-added chemicals or fuels, which can achieve sustainable goals [88,89]. Atomically dispersed mono-atomic catalysts are one of the most attractive electrocatalysts for the CO_2_ reduction reaction (ORR) [90]. As a new electrode material, a mono-atomic catalyst can cope with the challenges of slow dynamics of CO_2_RR and low Faraday efficiency. The single-atom catalytic system with higher reaction activity and higher selectivity under low overpotential can realize electrocatalytic CO_2_ reduction technology with higher energy efficiency and large-scale advantages. The precious metal gold and silver single-atom catalysts have a high Faraday efficiency in the CO_2_ reduction reaction, reaching more than 90%. Adjusting the coordination environment of single-atom catalysts and changing the electronic structure of the central atom can change the selectivity of the product and generate hydrocarbons other than CO [91]. The mono-atomic catalyst loaded on modified or doped carbon-based graphene shows superior electrochemical performance and can efficiently electrocatalyze CO_2_RR [92].

Jiao et al. [93] reported an atom-pair catalyst (APC) with the characteristics of two adjacent copper atoms. The catalyst has a highly active atomic interface. The active site of APC is composed of Cu1^0^-Cu1^x+^ pairs stabilized by Te surface defects of Pd_10_Te_3_ alloy nanowires. In the process of carbon dioxide reduction, the key bimolecular step is completed, Cu1^x+^ adsorbs H_2_O, and adjacent Cu1^0^ adsorbs CO_2_, thus promoting the activation of CO_2_. At low overpotential, the catalyst has high activity and Faraday efficiency in CO_2_RR to selectively produce CO (FeCO). Experimental characterization and density functional theory show that the adsorption configuration reduces the activation energy, and APC has high selectivity, activity, and stability at relatively low potential.

#### 5.1.6. Batteries

A mono-atomic catalyst has the characteristics of stability and high efficiency, which can be used in various batteries and electrochemical storage devices, can drive the research of a series of energy conversion and storage devices, and accelerate the exploration and development of green energy [94]. The highly stable non-precious metal M-NX/C single-atom catalyst has application prospects in proton-exchange-membrane fuel cells and promotes the development of the transportation industry [95].

Li et al. [96] loaded single Ag atoms directly on the surface of a-MnO_2_ nanorods to improve the oxygen reduction performance of Mg–air fuel cells. Modification of an Ag atom increases the content of Mn(III) and oxygen vacancy, and the catalytic activity of a-MnO_2_ is significantly improved. The modified silver content is very low, the half-wave potential of a-MnO_2_ (Ag@a-MnO_2_) is 0.85 V, the limiting current density is 5.8 mA cm^−2^, and the electron-transfer pathway is four electrons. It has good stability and anti-toxicity. The Mg–air fuel cell contains Ag@a-MnO_2_ and has a high power density of 107.2 mW cm^−2^, showing excellent discharge performance. This study provides a new idea for the preparation of efficient and low-cost ORR catalysts and the promotion of large-scale commercial production of metal–air fuel cells.

Wang et al. [97] fixed the central Mo atom of the oxygen and nitrogen distribution sites on the porous carbon by designing the coordination environment, and this unique molybdenum single-atom catalyst showed high ORR activity. Compared with the current Pt/C, the catalytic performance of ORR is outstanding under alkaline condition. Mo-O/N-C can also be used as an electrocatalyst and has remarkable performance as an air cathode in Zn–air batteries.

Xie et al. [98] prepared high-performance Co (MIM)–NC (1.0) single-atom catalysts by encapsulation ligand exchange using the immobilization of Con4 groups in ZIF-8 micropores. This has great potential in the field of proton-exchange-membrane fuel cells with high activity and durability. In oxygen reduction reaction, this synthesis method increased the density of active sites and enhanced the ORR activity of the catalyst. The current density and peak power density are larger. As an energy conversion device, a proton-exchange-membrane fuel cell (PEMFC) has the characteristics of high efficiency and cleaning, and requires a highly active catalyst for REDOX reactions at the cathode.

#### 5.1.7. Catalytic Reduction of NO_x_

Catalytic reduction of NO_x_ by H_2_ over metal catalysts has great prospects in controlling toxic gas emissions [99]. A single-atom alloy catalyst has synergistic effects and significantly enhances the activity of a single-atom catalyst, which can be used for efficient reduction of NO_x_ and shows excellent catalytic activity in NO_x_ reduction reaction.

Liu et al. [9] studied the mechanism of H_2_ catalytic reduction of N_2_O over SACS M1/PTA with different polyoxometalate-supported single-atom catalysts systematically. Among these, M1/PTA SAC and Os1/PTA 19 SAC showed high activity for H_2_ reduction of N_2_O and low rate-determination barrier. The effective catalytic pathways include the first and second degradation of N_2_O through Os1/PTA SAC and the hydrogenation of key substances after the second degradation of N_2_O. The molecular geometry and electronic structure of the effective reaction pathway were analyzed. The results showed that there was a strong charge transfer synergistic effect between the metal and the carrier, which effectively improved the catalytic activity of Os1/PTA SAC. The isolated Os atom has the function of adsorbing and activating N_2_O molecules, and acts as an electron-transfer medium during the entire reaction. The PTA carrier has high redox stability, can transfer electrons, and promote the entire reaction.

#### 5.1.8. Water–Gas Conversion Reaction

Water–gas conversion reaction provides an important way to produce hydrogen and purify carbon monoxide at the same time, and its combination with the steam reforming reaction is the main technology of cheap hydrogen production. However, the efficiency of traditional low-temperature water vapor transfer catalyst for hydrogen production is low, so the research and development of a new generation of low-temperature transfer catalyst has important scientific significance and practical value [100]. The activation energy of a gold catalyst in the water–gas shift reaction is relatively low. When alkali metal ions are added to the single-atom Au/molecular sieve catalyst, the coordination structure formed by Au and alkali metal ions can improve the stability of the active centers of dispersed atoms. The formation of Au-O(OH)x-(Na/K) clusters is more active in the water–gas shift reaction under low-temperature conditions [101].

Liang et al. [102] studied redox mechanism of Ir1/FeOx single-atom catalyst in water–gas conversion (WGS) reactions. Water is readily decomposed into OH^*^ on a single atom of IR1 and H^*^ on a nearby O atom bonded to the Fe position. The CO adsorbed on IR1 reacts with the adjacent O atom to produce CO_2_, resulting in oxygen vacancy (OVAC). H_2_ can be produced by the migration of H from the adsorbed OH^*^ to IR1 and subsequent reaction with another H^*^. The synergistic interaction of bimetal active sites formed between IR1 and neighboring Fe species, with the participation of surface oxygen vacancies, suggests that electrons transfer from the Fe^3+^-O∙∙∙Ir^2+^-OVac active site to the new site of Fe^2+^-OVac ∙∙∙Ir^3+^-O during the electron transition process. The REDOX mechanism of WGS reaction through synergistic bimetallic active sites (DMAS) is different from the traditional mechanism of the formation of formic acid or carboxyl intermediates. It is also revealed that the unique strong interaction between the covalent metal support can adjust the catalytic performance of the catalyst.

Sun et al. [103] reported a highly active Ir1/α-MoC single-atom catalyst for a low-temperature water–gas conversion reaction, with a CO conversion rate of 100% at 150 °C. The reaction rate of this catalyst is several orders of magnitude higher than that of other IR-based catalysts supported by other carriers, but is only about four times that of α-MOC. Experimental characterization combined with density functional (DFT) calculation revealed that the high activity of the catalyst was attributed to α-MOC support, while IR1 acted as a co-catalyst and did not directly participate in the reaction, changing the electronic structure of Mo active center and thus affecting the reactivity of a water–gas conversion.

### 5.2. Applications in the Field of Photocatalysis

#### 5.2.1. Photocatalytic Dehydrogenation (PER)

A variety of single-atom catalysts have demonstrated excellent breakthrough performance in the electrocatalysis of hydrogen evolution reactions in acidic and alkaline environments, which is expected to promote the large-scale applications of electrolysis of water for hydrogen production as a sustainable method of hydrogen production. Zhang et al. [104] reported a novel method for preparing single-atom catalysts—potential cyclic voltammetry. This method is used to synthesize an efficient single-atom catalyst for hydrogen evolution reaction—i.e., single-atom platinum supported on cobalt phosphide-based nanotube arrays. When it is applied to neutral hydrogen evolution reaction electrocatalysis, it exhibits better quality activity and stability than commercial Pt/C, and it also exhibits excellent performance in acidic and basic electrocatalytic hydrogen evolution reactions with the η value of PtSA-NT-NF at *j*_HER_ of 10mA cm^−2^ after the stability test, which shows a small increase of 8mV, much smaller than the increase of Pt/C, 128 mV.

Jin et al. [105] reported the construction of oxidized Ni single-atom materials on carbon nitride (CN), which achieved a 30-fold increase in photocatalytic hydrogen evolution activity. The study found that unpaired electrons in the 3D orbitals of partially oxidized Ni atoms are more likely to be excitation. This partially oxidized single-atom Ni/CN material exhibits high photoresponse performance, electrical conductivity and charge separation ability, and improved carrier performance. These features work together to achieve a substantial improvement in photocatalytic performance.

Li et al. [106] reported a single-atom Ag catalyst doped with g-C_3_N_4_ (SAAg-g-CN), as a low-cost and stable catalyst, has higher activity in photocatalytic hydrogen-desorbed reaction (PER) and solar–thermal-assisted PER process. The study found that the activity of SAAg-g-CN comes from the Gibbs free energy of the adsorbed hydrogen atoms and the firm N-Ag bond structure. The PER rate of SAAg-g-CN at 55 °C is twice that at 25 °C. Due to the agglomeration of metal NPs, the PER performance of AgNP-g-CN and PtNP-g-CN decreases. This study proved that SAAg-g-CN has ultrahigh photoactivity and photothermal stability, and has great potential in promoting full use of solar energy. This discovery provides a feasible strategy for using solar–thermal energy for efficient photocatalysis.

#### 5.2.2. Photocatalytic Water-Splitting

Solar energy photocatalytic water-splitting is expected to solve the energy and environmental crises facing the world, and has always attracted attention. Improving the conversion efficiency of solar energy to hydrogen energy has always been the focus and difficulty of this research field. Single-atom co-catalysts have shown great potential in the application of photocatalytic water-splitting to produce hydrogen. On the one hand, reducing the particle size of the co-catalyst to the atomic level can maximize the catalytic reaction sites; on the other hand, the specific coordination between the single-atom co-catalyst and the semiconductor photocatalyst may produce unique electrical properties, therefore promoting the progress of the catalytic reaction.

Zuo et al. [107] established a new method of surfactant-assisted self-assembly, selecting organic ligands containing Pt porphyrin and copper ions as the building units of MOF. With the assistance of PVP surfactant, ultrathin porphyrin-based MOF nanosheets with ultrahigh Pt mono-atom loading were prepared from bottom to top through the coordination of carboxyl groups and copper ions. This material has significant advantages in the photocatalytic process. The ultrathin thickness and high specific surface area fully expose many active sites, while shortening the migration distance of photogenerated carriers from the inside of the material to the surface, greatly reducing the probability of recombination quenching of photogenerated carriers. In the process of photocatalytic water-splitting and hydrogen production under visible light irradiation, PtSA-MNS exhibits excellent hydrogen production performance and cycle stability. Nanosheets can also be deposited onto solid substrates through a simple solvent evaporation film formation method to form a uniform film, which retains high photocatalytic activity. The results show that PTSA-MNS has good machinability.

#### 5.2.3. CO_2_ Reduction

Han et al. [108] used photocatalytic technology, driven by solar energy, where the low-concentration CO_2_ contained in industrial waste gas is directly reduced to chemical raw materials or carbon-based fuel by photocatalysis. This technology provides a potential solution for reducing CO_2_ emissions and producing value-added products to realize resource use and alleviate global climate change. However, due to the high stability of CO_2_ molecules and the complex multi-electron reaction process, efficient and selective CO_2_ photocatalytic reduction is still a huge challenge.

Ji et al. [109] proposed an atomic confinement and coordination strategy to prepare a rare-earth erbium single-atom (Er_1_/CN-NT) catalyst with a controllable dispersion density supported on carbon nitride nanotubes. In situ XAFS and DFT calculations together show that the uniform dispersion of erbium at the atomic level has a major contribution to the excellent performance of photocatalytic CO_2_RR.

Wang et al. [56] used light-induced methods to anchor Cu single atoms (Cu SAs/UiO-66-NH_2_) on the UiO-66-NH_2_ carrier that can significantly promote the conversion of CO_2_ into liquid fuel. Driven by sunlight, the prepared Cu SAs/UiO-66-NH_2_ photocatalytic conversion rates of CO_2_ to methanol and ethanol are 5.33 and 4.22 μmol h^−1^g^−1^, respectively. The yield is much higher than that of the initial UiO-66-NH_2_ and Cu NPs/UiO-66-NH_2_ complex. Density functional theory (DFT) calculations show that the introduction of Cu SAs on UiO-66-NH_2_ can greatly promote the conversion of CO_2_ to CHO* and CO* intermediates, therefore producing methanol and ethanol with extremely high selectivity.

#### 5.2.4. Photocatalytic Nitrogen Fixation

Photocatalytic technology can directly convert solar energy into chemical energy, which is excited by light, and an electron-transfer reaction occurs, and the electron is transferred to the ultra-stable nitrogen–nitrogen bond. It provides a very promising method for reducing the energy consumption of synthetic ammonia. However, the ultrahigh bond energy of the nitrogen–nitrogen bond makes the nitrogen molecule exhibit stable chemical characteristics, which makes it difficult for conventional photocatalytic materials to activate the nitrogen molecule. Therefore, it is necessary to develop an efficient nitrogen-fixing ammonia synthesis photocatalyst [110].

Liu et al. [111] successfully prepared a new type of phosphorus-doped Bi_2_WO_6_ (PBWO) single-atom photocatalyst by a simple and easy one-step hydrothermal method, which has abundant oxygen vacancies. P atoms partially replace Bi atoms, forming Bi-O-P bonds in the [BiO]^+^ layer and then doping into the Bi_2_WO_6_ lattice. The synthesized catalyst samples have significant catalytic activity. Under visible light irradiation, the highest photocatalytic efficiency of water detoxication was achieved by PBWO-40. The corresponding kinetic rate constant for Cr(VI) reduction is calculated as 0.0918 min^−1^, which is 2.99 times, 1.65 times, and 1.29 times higher than that of BWO, BPO/BWO-20, and RP/BWO-30. It has a better degradation effect on Cr(VI) and tetracycline hydrochloride. Under simulated light irradiation, it has strong nitrogen fixation activity and good stability. Compared with other phosphorus-containing heterostructure catalysts, the photocatalytic nitrogen fixation rate of the catalyst is significantly improved. According to various characterization and DFT calculation results, the synergistic effect between phosphorus doping and oxygen vacancies gives PBWO excellent photocatalytic activity. Therefore, the absorption of visible light is enhanced, the energy band is narrowed, and the defect level is introduced in the forbidden band, which promotes the high separation efficiency of photogenerated electron-hole pairs.

Wu et al. [112] prepared a three-dimensional (3D) mono-atomic iron-doped large/mesoporous TiO_2_-SiO_2_ (Fe-T-S) photocatalyst using the evaporation-induced self-assembly (EISA) method. The effect of Fe modification on the whole ammonia formation process was also studied. The experimental results show that the ammonia synthesis rate of the catalyst is as high as 32 μmol g^−1^ h^−1^ without any sacrificial agent and co-catalyst, which is 4.25 times that of the undoped Fe sample. Fe doping promotes the generation of O_2_ and inhibits the amount of hydrogen evolution. Experimental results and density functional theory (DFT) calculations show that the modification of Fe leads to the formation of photoinduced hole capture polarity, and then Fe exists as high iron (IV). The single-atom Fe(IV) site plays a leading role in the process of water oxidation and helps to promote the reduction of nitrogen on the adjacent oxygen vacancy.

### 5.3. Sensing

The application of single-atom catalysts in the electrochemical field was expanded using a single-atom catalyst to construct a sensing interface for electrochemical analysis. By effectively linking atomic and electronic structures with macroscopic electrochemical phenomena, it has been demonstrated that the electrocatalytic ability of single-atom catalysts as sensing materials plays a crucial role in the sensitive and selective detection of analytes [113].

Li et al. [114] designed sensor materials with isolated single-atom sites of precious metals. The one-dimensionally arranged porous γ-Fe_2_O_3_ nanoparticle composite material is doped with atom-dispersed platinum as an ethanol gas sensor, which has high selectivity to ethanol. The Pt single-atom site has a high valence state, which can enhance the adsorption capacity of the carrier to ethanol molecules. The improvement of the sensor sensitivity is mainly through the single-atom site changing the electronic structure of the carrier. It can maximize the efficiency of metal atoms and reduce application costs.

Single-atom catalysts exhibit good electrical conductivity and strong electrocatalytic activity, and can be used in electrochemical biosensing analysis. Single-atom catalysts have better signal amplification effects than traditional nano-electrocatalysts. By optimizing the adsorption method of the single metal site to the detection target, the sensitivity of electrochemical detection of the target can be enhanced. In addition, in the electrochemical detection of H_2_O_2_, single-atom catalysts have higher reduction potential and reduction current than traditional electrocatalysts. The single-atom catalyst has a unique catalytic activity center, which has high selectivity to H_2_O_2_ and O_2_, and can realize the sensitive detection of O_2_ in vivo [115].

### 5.4. Degradation of Organic Pollutants

The current degradation methods of organic pollutants in water mainly include direct catalytic oxidation, photocatalytic oxidation, new advanced oxidation based on activated persulfate [116], and photoelectric catalytic oxidation [117]. Direct oxidation technology is divided into the Fenton method and ozone advanced oxidation method. The development of modified catalysts with excellent catalytic activity and stability to achieve efficient degradation of organic pollutants is still the focus of practical research [118,119].

Chen et al. [120] prepared efficient Fe/O co-doped g-C_3_N_4_ nanosheet catalyst by a simple calcination method, and the degradation mechanism of the catalyst on organic pollutants was clarified. In contrast to the traditional advanced oxidation technology based on PMS, the activation of PMS is through singlet oxygen and Fe(V)=O double nonradical pathway. Adding lower doses of Fe/O-doped g-C_3_N_4_ and PMS to the simulated high-salt wastewater can efficiently and rapidly degrade the model pollutant bisphenol a, with a large reaction rate constant. The co-modification of O and Fe atoms can change the electronic structure of g-C_3_N_4_, promote the rapid transfer of electrons, and generate more highly active nonradical species. The results of electrochemical experiments and density functional theory calculations revealed that Fe(V)=O played a key role in the degradation of BPA and promoted the transfer of electrons from BPA molecules to the PMS/catalyst complex. The catalytic system is suitable for a wide pH range and a variety of organic pollutants, and has broad application prospects.

Qi et al. [121] used lignin as a carbon carrier. A one-pot pyrolysis strategy was used to prepare a nitrogen-coordinated Co single-atom catalyst (SACo-N/C). The HAADF-STEM image and XAS analysis show that in the entire SACo-N/C structure, the Co atom dispersion is relatively high. The catalytic activity of the catalyst is significantly improved after PMS activation. Studies have found that the main active site for PMS activation and degradation of NPX is the single-atom Co site. After repeated use of the catalyst, it was found that its catalytic activity was not significantly reduced. The degradation mechanism of NPX is studied, and the results showed that the electron transfer was caused by a single-atom Co site.

Pan et al. [122] constructed a mono-atomic copper catalyst (SACu@NBC) using a biogas residue-based adsorbent saturated with copper and zinc ions as a raw material, mixed with dicyandiamide and pyrolyzed, and used to activate peroxymonosulfate (PMS) degrade organic pollutants in water bodies. The degradation effects of 6 phenolic pollutants in the 3SACu@NBC/PMS system were tested, revealing the oxidation mechanism dominated by electron transfer in the system. The results show that with the introduction of mono-atomic copper, the chemical reaction activity of the catalyst is significantly improved, and the charge transfer ability is further enhanced. The quenching experiment, FFA decomposition experiment and EPR results show that the activation of PMS in the system is not dominated by free radicals, and 1O_2_ is not the main active substance in the 3SACu@NBC/PMS system.

## 6. Conclusions and Prospects

Presently, researchers have developed a variety of strategies to synthesize different kinds of mono-atomic catalysts, and we can also observe the morphological structure of mono-atomic catalysts with the help of advanced characterization technology, and explore their physical and chemical properties in depth. The development and evolution of mono-atomic catalysts move catalytic reactions from the macro field to the micro world. It provides an ideal model and a broad research platform for people to understand catalytic reactions at an atomic scale.

One of the forefront topics in SACs is the high concentration of metal atoms loaded. The high concentration loaded SACs can improve surface area and effective mass activity which is a prospect for application. Due to the surface energy, it is quite easy to aggregate individual atoms nearby. Therefore, a series of methods such as atomic layer deposition, co-precipitation method, dipping method, low-temperature photoreduction method, flame-spray pyrolysis method, high-temperature cracking method, underpotential deposition method, ball-milling method, or molten salt method have been focused on to challenge the well-dispersed high-loading metal atoms. Additionally, we can divide the methods by physical deposition, which have a low metal loading, thermal volatile molecules to capture those atoms easily aggregated with high temperatures, and pyrolysis and thermal hydrolysis that would control the synthesis conveniently.

Secondly, we must avoid high surface energy to disperse metal atoms on the support. Under this condition, an appropriate substrate to form strong interaction with single metal atoms is necessary. Using different-dimension substrates, high-loading metal atoms could be prepared, which demonstrate high activity and selectivity. Because of the high specific surface area, the SACs could make more accessible interactions with atom active sites that are exposed on the supporting surface. By the strong interaction of doping, defects, and vacancies with single metal atoms and support, forming heteroatoms can improve the metal atom immobilization.

However, SACs promise to be the ideal catalysts for maximum atomic use. Although it can use metal resources and promote electrocatalytic systems rationally, limitations are the shortages that must improve the loading of metal single atoms. Recently, the loading value has been enhanced, but there is still huge room for improving atom efficiency, selectivity. Therefore, the opportunities and challenges in the practical application of SACs are:

(1) Industrial applications usually require a higher loading of single-atom catalysts. It is of great significance to increase the loading of single-atom catalytic materials and achieve large-scale preparation, and give full play to the application potential of single-atom catalysts in the industrial field. Using regular methods it is hard to improve single metal atom loading continuously, especially for large-scale synthesis. New strategies are urgently required for further applications.

(2) The coordination structure of the metal catalyst should be adjusted rationally, and the metal single atom should be stabilized through the charge transfer effect between the coordination atoms around it, and a highly active single-atom catalyst with the best electronic structure should be designed. The precursors are quite important for mononuclear metal complexes. To improve highly dispersed single metal atoms, the spatial confinement, trapping by defect or vacancy, ligand anchoring, and low-temperature inhibition of molecular thermal motion etc. could be considered. The stability of SACs relies on the strong metal–support interactions or robust chemical bonds which concerns the relationship between single metal atoms and surrounding coordination atoms. The active sites should be maximized to make mass transportation easy by designing highly specific surface-area supports.

(3) SACs are ideal catalytical systems that can be rationally designed, but there are still lots of risks in industrial applications. We must do more research on theoretical calculation, to clearly specify the real active sites and gain insight into the relationship between structure and activity. Chemical reaction pathways and mechanisms should be demonstrated.

(4) Alloy metal atoms could improve metal atom loading, catalytic efficiency, and selectivity, and are highly dispersed. The alloy metal atoms catalysts can give several metals a symbiotic role, which effectively catalyzes the reactions. The DFT calculations showed that either homonuclear or heteronuclear could attract catalytic performance. Although the synthesis of single-alloy–atom catalysts is in its infancy, great potential for certain applications have been discovered. Increasing numbers of single-alloy–atom catalysts are deeply understood, with detailed design and rational synthesis.

In this way, highly efficient catalysts could be rationally produced for atomic chemical reactions, which could maximize the potential of atoms and make green catalysis more cost-effective.

## Figures and Tables

**Figure 1 molecules-26-06501-f001:**
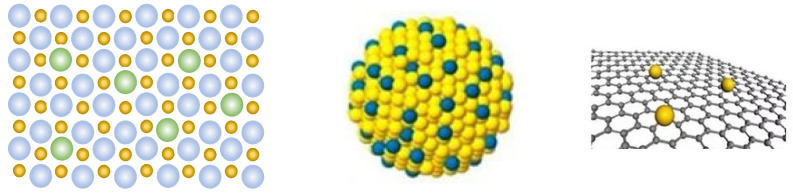
SAC structure model.

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
