# Peer review of "Research Progress and Application of Single-Atom Catalysts: A Review"

_molecules, 2021, doi:10.3390/molecules26216501_

Round 1

Reviewer 1 Report

The manuscript is interesting and brings worth data for the community.

It can be accepted in its present form. 

The authors are requested to verify once mnore spelling and English.

Author Response

We appreciate your warm work earnestly and your comments. Thank you very much for your approval.

Reviewer 2 Report

Dear Authors,

Thank you for delivering the review article. But in my opinion, in this work should be explained the mechanism of catalysis on this kind of materials.

Besides, I have a few other comments:

  • There is some mess with acronyms, e.g. "SAC" is not explained, "ALD" is repeated (line 196), "SAA" in line 279
  • correct the references in the text, e.g. in line 42, 86
  • I miss pictures showing, for example, the atomic structure of different types of catalysts
  • In general, there are a lot of terms in the text like "better", "higher", "excellent" etc., it is not precise. I suggest the authors clarify it by specific values.
  • In chapter 3.1. I suggest to write what kind of parameters can be controlled in the ALD method
  • I suggest to create a summary table of catalysts produced by various methods, it will certainly make it easier for the reader to find interesting information.
  • There are two examples of works for each synthesis method, I think it is not enough for review arcitle.
  • Chapter 4 - how does morphology affect properties? There is no explanation.
  • There are typos in the text, please check it carefully next time.

Author Response

Response to Reviewer 2 Comments Point 1: There is some mess with acronyms, e.g. "SAC" is not explained, "ALD" is repeated (line 196), "SAA" in line 279 Response 1: SAC had been mentioned in L20 which is “Single-atom catalyst”. The ALD is a method to prepare catalyst which is Atomic layer deposition. The SAA in line 279 means a kind of catalyst which is single atom alloy catalyst. Line 279 Pd1Ag3/Al2O3 monoatomic alloy catalyst is a kind of single atom allyoy catalyst. So we revised this sentence as Rassolov, A.V et al. synthesized the monoatomic alloy Pd-Ag catalyst by wet impregnation method, and studied the catalytic performance of the Pd1Ag3/Al2O3 monoatomic alloy catalyst, a kind of SAA catalyst, in the liquid phase hydrogenation of diphenylacetylene. Point 2: correct the references in the text, e.g. in line 42, 86 Response 2: We had check the references in the text and modified the text. We had summarized a title for 2.2.1.2, and the contents had modified. . Point 3: I miss pictures showing, for example, the atomic structure of different types of catalysts. Response 3: We had added 3 pictures in the first chapter. Point 4: In general, there are a lot of terms in the text like "better", "higher", "excellent" etc., it is not precise. I suggest the authors clarify it by specific values.. Response 4: We had revised the text to clarify the terms such as "better", "higher", "excellent". Point 5: In chapter 3.1. I suggest to write what kind of parameters can be controlled in the ALD method Response 5: We had summarized the key parameters controlled in ALD method which were the precursor as template and the exposure time for the precursor. Point 6: There are two examples of works for each synthesis method, I think it is not enough for review arcitle. Response 6: Each synthesis method has a regular method and for the detailed catalyst there may be tiny differences to control its shape such as exposure time, precursors etc. Herein we gave a general method for the researcher for reference. Point 7: Chapter 4 - how does morphology affect properties? There is no explanation. Response 7: In chapter 4 we were discussing the regular characterization technology of single atom catalyst. The morphology can affect properties when the catalyst was used in the reactions which depended on the chemical reactions. Difference reactions had suitable parameters including morphology. So we do not discuss the affection of morphology in this paper. Point 8: There are typos in the text, please check it carefully next time. Response 8: We have check the manuscript again. We appreciate your warm work earnestly and hope that the correction will meet with approval. Once again, thank you very much for your comments and suggestions.

Reviewer 3 Report

In this review, He et al. summarize and discuss the current research progress on single atom catalysts. The author covered a wide range of topic and provide detailed example. However, I have some concern about this manuscript. I would only recommend this review to be published on Molecules If the author can address my concerns.

1) The introduction is too short. It is more like an section of abstract. There should be solid discussion on the theory and history and single-atom catalysts.

2) There are too many application of SAC that the author would like to cover. For example, in section 5.1 Applications in the field of electrochemistry, eight different sub-sections were listed. It makes this review like a list of short summary of different literature results. The major problem is in-depth discussion is missing.

3) The Conclusions and Prospects is again too short. The author need to ellaborate their own opinion and suggestion to this field instead of providing some bullet points.

Author Response

Response to Reviewer 3 Comments Point 1: The introduction is too short. It is more like an section of abstract. There should be solid discussion on the theory and history and single-atom catalysts. Response 1: We are very sorry for our negligence of solid discussion on the theory and history and single-atom catalysts. This comment is valuable and very helpful for revising and improving our paper, as well as the important guiding significance to our researches. We have studied comments carefully and have made corrections which we hope meet with approval. We further summarized the development history of single-atom catalysts and discussed its theory in depth, which marked in red in the paper. Point 2: There are too many application of SAC that the author would like to cover. For example, in section 5.1 Applications in the field of electrochemistry, eight different sub-sections were listed. It makes this review like a list of short summary of different literature results. The major problem is in-depth discussion is missing. Response 2: We have studied reviewer’s comments carefully and have made revision which marked in red in the paper. We have tried our best to revise our manuscript according to the comments. Attached please find the revised version, which we would like to submit for your kind consideration. Point 3: The Conclusions and Prospects is again too short. The author need to ellaborate their own opinion and suggestion to this field instead of providing some bullet points. Response 3: We have revised the Conclusions and Prospects by our opinions and give some suggestion to SACs.

Round 2

Reviewer 2 Report

Dear authors,

Thank you for including my comments in the text correction. I think the manuscript can be accepted for publication. 

Reviewer 3 Report

I highly appreciate the authors' effort on addressing my concern about this review. This review article should be able to provide the field a great resource to study the background and developement of single atom catalysts. I would like to recommend this manuscript to be published on Molecules.